# Effect of Cr Atom Plasma Emission Intensity on the Characteristics of Cr-DLC Films Deposited by Pulsed-DC Magnetron Sputtering

**Guang Li** [1,2,†]**, Yi Xu** [1,2,†] **and Yuan Xia** [1,2,*]

1   Institute of Mechanics, Chinese Academy of Sciences, Beijing 100190, China; lghit@imech.ac.cn (G.L.); xuyi@imech.ac.cn (Y.X.)
2   Center of Materials Science and Optoelectronics Engineering, University of Chinese Academy of Sciences, Beijing 100049, China
*   Correspondence: xia@imech.ac.cn
†   These authors contributed equally to this work.

**Abstract:** A pulsed-dc (direct current) magnetron sputtering with a plasma emission monitor (PEM) system was applied to synthesize Cr-containing hydrogenated amorphous diamond-like carbon (Cr-DLC) films using a large-size industrial Cr target. The plasma emission intensity of a Cr atom at 358 nm wavelength was characterized by optical emission spectrometer (OES). $C_2H_2$ gas flow rate was precisely adjusted to obtain a stable plasma emission intensity. The relationships between Cr atom plasma emission intensity and the element concentration, cross-sectional morphology, deposition rate, microstructure, mechanical properties, and tribological properties of Cr-DLC films were investigated. Scanning electron microscope and Raman spectra were employed to analyze the chemical composition and microstructure, respectively. The mechanical and tribological behaviors were characterized and analyzed by using the nano-indentation, scratch test instrument, and ball-on-disk reciprocating friction/wear tester. The results indicate that the PEM system was successfully used in magnetron sputtering for a more stable Cr-DLC deposition process.

**Keywords:** diamond-like carbon (DLC); plasma emission intensity; magnetron sputtering; tribological performance

## 1. Introduction

Metal-containing hydrogenated amorphous diamond-like carbon films (Me-DLC) have raised much attention for decades because of their promising performance, such as high nano-hardness, chemical inertness, excellent tribological properties, and thermal stability [1–4]. By the doping of metals, a two-dimensional array of nanoclusters within the DLC matrix or an atomic-scale composite can be formed, and these nanostructures are responsible for the desirable mechanical and thermal properties [5–7]. Generally, Me-DLC films are synthesized by hybrid magnetron sputtering using a metal target with a hydrocarbon gas [8,9]. However, an inherent feature in this conventional hybrid deposition process, combining magnetron and plasma assisted chemical vapor deposition, is high instability due to the complex relation between the fluxes of reactive gas and sputtered metal from the target. During the deposition process, it inevitably suffers from the "hysteresis effect", immediate rapid transition from metal to poisoned state [10,11]. Such behavior originates from the formation of carbide on the target surface, leading to a significant decrease in the sputtering yield or a non-reproducible quality of films. Therefore, precise control of the 'poisoned' state of the metal target surface is vital for the deposition of high quality Me-DLC films.

Plasma emission monitoring (PEM) system, based on the gas flow control technique, has received extensive concern as an important method of stabilizing the deposition process in the "transition" state because of its high accuracy and stability [12,13]. Wu et al. showed that various $CrN_X$ films were obtained under a stable deposition mode by adjusting the $N_2$ gas flow rate and exhibiting different Cr atom plasma emission intensities [14]. Ohno et al. stated that stable depositions for $TiO_2$ films were achieved in the transition region with the PEM system [15]. Hirohata et al. reported the sputtering of ZnAl in $O_2$ atmosphere with the PEM system and showed that ZnO films deposition rates were about 15 times higher than those by conventional magnetron sputter [16]. In a similar way, Yang et al. investigated the properties of Ti-containing and W-containing hydrogenated DLC films deposited via high power impulse magnetron sputtering (HiPIMS), by controlling the $C_2H_2$ flow rate and different $Ti^{2+}$ plasma emission intensities under the PEM system [17,18]. They found that the hysteresis curve depicted almost completely overlapping curves with some fluctuations in the amount of $C_2H_2$, which implied that the PEM controlled HiPIMS process was easy, accurate, and stable, especially in reactive poisoning deposition. As can be clearly seen from these publications, in comparison to conventional hybrid magnetron sputtering, the PEM system exhibits excellent capabilities for improving film deposition rate and stabilizing the deposition process. On the other hand, however, the sputtered targets in most of these works are small in size. For example, 200 mm-diameter Fe and Zr targets were chosen in the publications [12,13], respectively. Only a few of them concern the deposition of films using a large-sized target under the PEM system. Compared with the small sized targets, the large sized targets are more suitable for industrial production. Due to the larger sputtered area, the target poisoning and abnormal discharge (arcing) phenomenon for large sized targets were more serious, which resulted in an unstable process or a nonreproducible quality of thin films for industrial application.

In this work, a pulsed-dc magnetron sputtering with PEM system was applied to fabricate Cr-DLC films using a large-size industrial Cr target, and $C_2H_2$ as feeding gas. The synthesis and characteristics were examined to evaluate the feasibility of a PEM system. To this end, the stability of Cr-DLC film deposition process with the PEM system was compared to that of conventional hybrid magnetron sputtering. The effect of Cr atom plasma emission intensity on the element concentration, cross-sectional morphology, deposition rate, microstructure, mechanical properties, and tribological properties of Cr-DLC films were explored. Our results demonstrate that the PEM system is an effective method for stabilizing the deposition process of the Cr-DLC films at different Cr atom plasma emission intensities, and the Cr-DLC film deposition rate is as high as 92.7 nm/min.

## 2. Experiment Details

### 2.1. Deposition

As shown in Figure 1, the experiments were conducted with an unbalanced magnetron sputtering system. A closed-loop plasma emission monitor system (PEM, Nava Fabrica FlotronTM, Vilnius, Lithuania) was utilized to precisely control the $C_2H_2$ gas flow rate. The plasma emission intensity of the Cr atom at 358 nm (see Figure 2) was characterized by a collimator of optical emission spectrometer (OES, Fabrica Flotron TM, Vilnius, Lithuania), which was placed 50 mm away from the target. During the deposition process, a pulsed DC power (MSP-20D, Pulse Tech, Nanjing, China) was used to power the Cr target, while a DC power (MSP-50D, Pulse Tech, Nanjing, China) was applied to the substrate. In this work, the set values of 100% and 0% were defined as the pure metallic deposition and fully poisoning deposition, respectively. The set values between 0% and 100% represent the percentage of the reduced Cr atom plasma emission intensity.

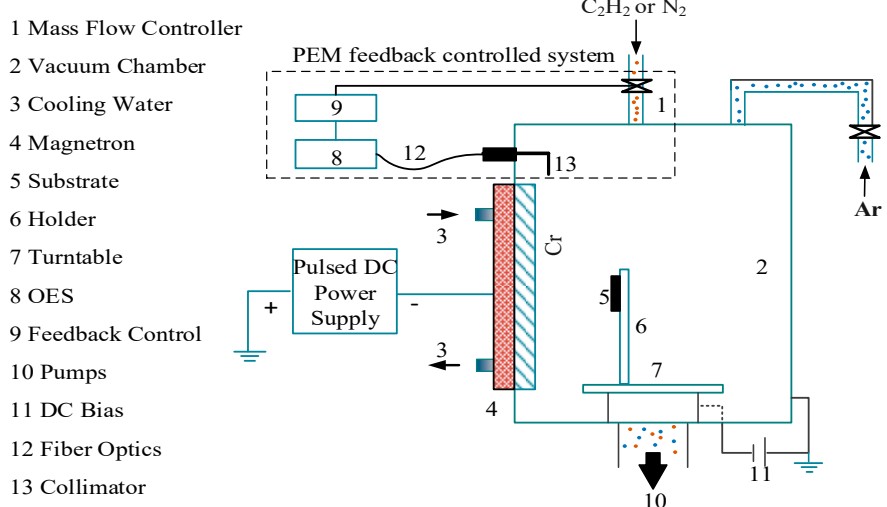

**Figure 1.** Schematics of deposition apparatus for Cr-containing hydrogenated amorphous diamond-like carbon (Cr-DLC) films.

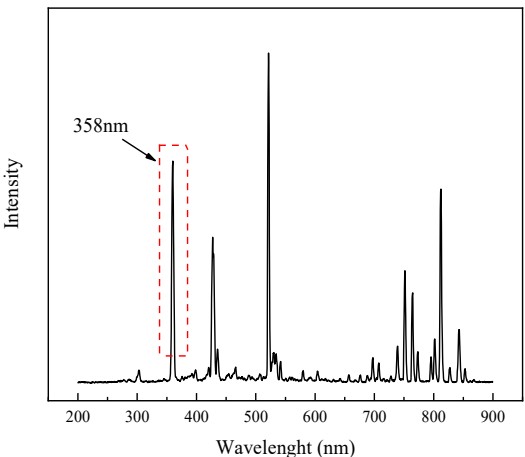

**Figure 2.** Typical optical emission spectrometer (OES) spectra during sputtering Cr target powered by pulsed-dc (direct current) magnetron sputtering.

There were two kinds of samples mounted in the vacuum chamber: high speed steel (HSS) samples ($\phi$ 30 mm) and silicon wafer ($5 \times 5$ mm$^2$). The vacuum chamber was mounted vertically with rectangular Cr (99.99% purity) target ($131 \times 580$ mm$^2$). The base vacuum of the chamber was less than $5 \times 10^{-3}$ Pa. After being ultrasonically pre-cleaned, all samples were further cleaned for 20 min in an Ar plasma environment (120 sccm, 2 Pa, $-900$ V bias voltage) in the chamber to remove surface contaminants. Then, the deposition of the layered interlayer containing Cr (150 nm) and CrN (500 nm) were deposited at a bias of $-75$ V in sequence at an Ar gas deposition pressure of 0.5 Pa in a pulsed DC mode to power the target to 5 A with a frequency of 40 kHz and a duty ratio of 80%. The CrN films were deposited under the PEM system, and the set value was fixed at 60%. The Cr/CrN interlayers were pre-deposited in order to improve the adhesion of the Cr-DLC films to the substrates. The deposition parameters for the Cr-DLC films are listed in detail in Table 1. During the deposition process, the substrate holder is stationary, and no additional heading was applied. For all deposition conditions, the substrate temperature was below 75 °C.

**Table 1.** The deposition parameters of Cr-DLC films.

| Parameter | Value |
|---|---|
| Target-Substrate Distance (mm) | 85 |
| Ar Gas Pressure (Pa) | 0.5 |
| Bias Voltage (V) | −75 |
| Set Value (%) | 2/5/10/15/20 |
| Pulse Frequency (kHz) | 40 |
| Duty Ratio (%) | 80 |
| Pulse Current (A) | 5 |
| Average power (kW) | 1.44/1.42/1.38/1.36/1.33 (stable deposition state) |
| Deposition Time (min) | 50 |

## 2.2. Characterization

A scanning electron microscope (FE-SEM, XL30 S-FEG, Philips, Amsterdam, Netherlands) with energy dispersive spectrometer (EDS, Philips, Amsterdam, Netherlands) was used to obtain the chemical composition and cross-sectional morphology. Raman spectra were analyzed by using an excitation wavelength of 523 nm and an argon ion laser, to characterize the chemical substructures of Cr-DLC films. The nano-hardness and modulus were investigated by a nano-indentation tester (Nano-Indentor G200, Agilent, SC, USA) with a load precision of 50 nN in continuous stiffness method, and an indentation of 300 nm was selected to avoid the substrate effect. A scratch test instrument (MFT-R4000, Huahui, Lanzhou, China) was employed to study the adhesion strength. The load was gradually increased from 0 to 50 N with 5 mm/min at a loading speed of 100 N/min. The friction coefficients and wear rates were evaluated by a ball-on-disk reciprocating friction/wear tester (MFT-R4000, Huahui, Lanzhou, China) at a relative humidity of approximately 22% RH and temperature of 23 °C. GCr15 bearing steel balls 5mm in diameter were used against the films at 15 N ($F$) and a sliding speed of 240 mm/min ($V$). The track length ($D$) and the duration time ($T$) were 5 mm and 60 min, respectively. After the wear experiments, the profiles of the tracks were measured by a surface profiler (MFT-R4000, Huahui, Lanzhou, China), and the cross-sectional areas of the wear tracks ($A$) were measured on each wear track at three points. The wear rates of the Cr-DLC films were calculated as follows: $(A \times D)/(F \times V \times T)$.

## 3. Results and Discussion

### 3.1. Stability Comparison: Constant Flow Control Mode vs. PEM System

A typical set value-time relationship for Cr-DLC depositing under constant flow control mode (53 sccm) is plotted in Figure 3a. The set value of Cr is about 12.4% in the initial stage. With the increase in time, the set value of Cr dramatically decreases, and then gradually approaches about 6% at 1100 s. For times greater than 1100 s, the set value of Cr kept a relatively stable value. The set value of Cr at 1100 s is nearly two times lower than that at the beginning. These phenomena can be attributed to the formation of carbide on the target surface, which reduces the number of sputtered particles from the Cr targets, thereby leading to the "target poisoning" effect [19,20]. Therefore, it is obvious that the constant flow control mode does not have the capacity for offering a stable deposition process. Here, PEM system offers a precise control and stable operation of the deposition process in Figure 3b. As the deposition of Cr-DLC films starts, the set value of Cr decreases quickly to the set value and then stays stable. By automatically decreasing the $C_2H_2$ gas flow rate from 61 to 53 sccm, the set value remains a stable state throughout the following deposition process. This dynamic process can be understood as follows: sputtering of the carbide on the Cr target is increased by an automatic decrease of the $C_2H_2$ flow rate in the PEM system. Subsequently, a stable Cr emission intensity can be maintained, which results in the Cr target surface being stabilized in the "transition" state, rather than 'poisoned' state. The target voltages and pressures during the deposition process under the PEM system at different set values are shown in Figure 4a,b, respectively. Like the set value of Cr above, the target voltage

and the deposition pressure both remain stable and no obvious drift occurred to either side during the deposition process, except for a very short period of adjustment at the beginning. Meanwhile, as the pulse current is kept constant (5A), the average power has the same variation tendency to the target voltage, and the average powers at the stable deposition state are 1.44 kW at 2%, 1.42 kW at 5%, 1.38 kW at 10%, 1.36 kW at 15%, and 1.33 kW at 20%. These results further indicate that the PEM system is an effective method for stabilizing the deposition process of the Cr-DLC films.

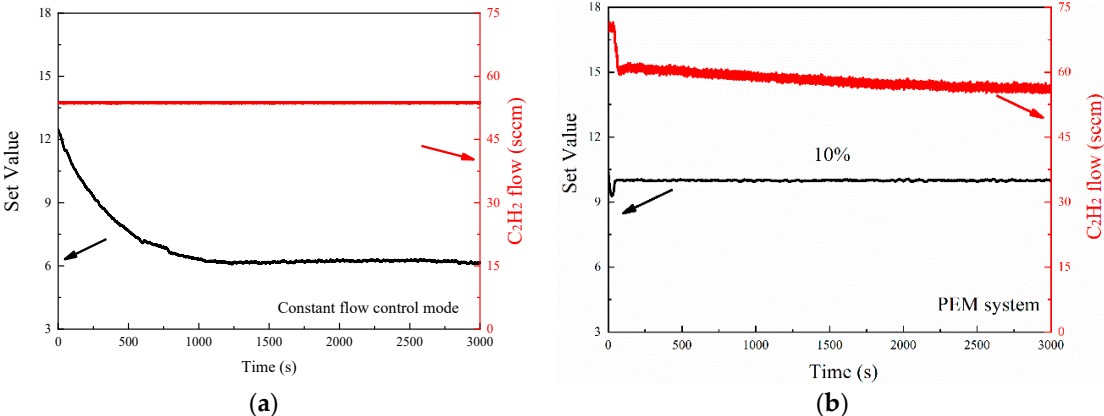

**Figure 3.** Experimental time-dependent emission intensity of Cr atom and $C_2H_2$ flow rate under (**a**) constant flow control mode and (**b**) plasma emission monitoring (PEM) system at a set value of 10%. The 0 s is the time of setting the $C_2H_2$ gas flow rate for constant flow control and set value for the PEM system.

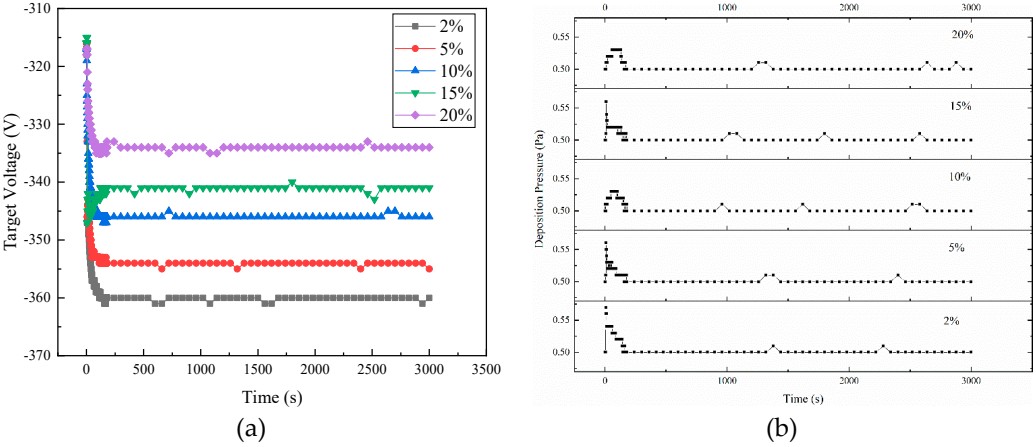

**Figure 4.** Experimental time-dependent (**a**) target voltages and (**b**) deposition pressures under the PEM system at different set values.

### 3.2. Effect of Set Values on the Composition and Deposition Rate

Table 2 lists the chemical composition of the Cr-DLC films deposited at different set values. Due to the fact that the hydrogen could not be detected by EDS, the relative atomic composition ratio was considered a sum of chromium, carbon, and oxygen. The relative content of C decreased from 97.59 to 82.34 at.% as the set value increased from 2% to 20%, which was likely due to the lower amount of $C_2H_2$ gas that was introduced. The Cr relative content increased from 1.96 to 17.21 at.%, with a decreased degree of target poisoning. In addition, there was a small amount of oxygen in the films, which may be caused by the contamination from the residual vacuum.

**Table 2.** Composition of Cr-DLC films.

| Set Value | Element Composition (at.%) | | |
|---|---|---|---|
| | C | Cr | O |
| 2% | 97.59 | 1.96 | 0.45 |
| 5% | 93.88 | 5.78 | 0.34 |
| 10% | 89.87 | 9.26 | 0.87 |
| 15% | 86.11 | 13.61 | 0.28 |
| 20% | 82.34 | 17.21 | 0.45 |

The cross-sectional morphologies of Cr-DLC films deposited at different set values are illustrated in Figure 5. It can be seen that the films all exhibit a typical dense morphology. According to the images in Figure 5, it was observed that the film thicknesses at the set values of 2%, 5%, 10%, 15%, and 20% were 4.6, 3.8, 3.6, 3.3 and 3.1 μm, respectively. The deposition rates for Cr-DLC films were obtained by dividing the thickness with the deposition time, and further exhibited in Figure 6. The film deposition rate exhibits a negative relationship with the set value. When the set value was 2%, the Cr-DLC film deposition rate was as high as 92.7 nm/min. Although the deposition rate decreased to 62.9 μm/h at a set value of 20%, the deposition of these Cr-DLC films under the PEM system presented a higher film deposition rate than that of conventional hybrid magnetron sputter deposition processes [9,21–24].

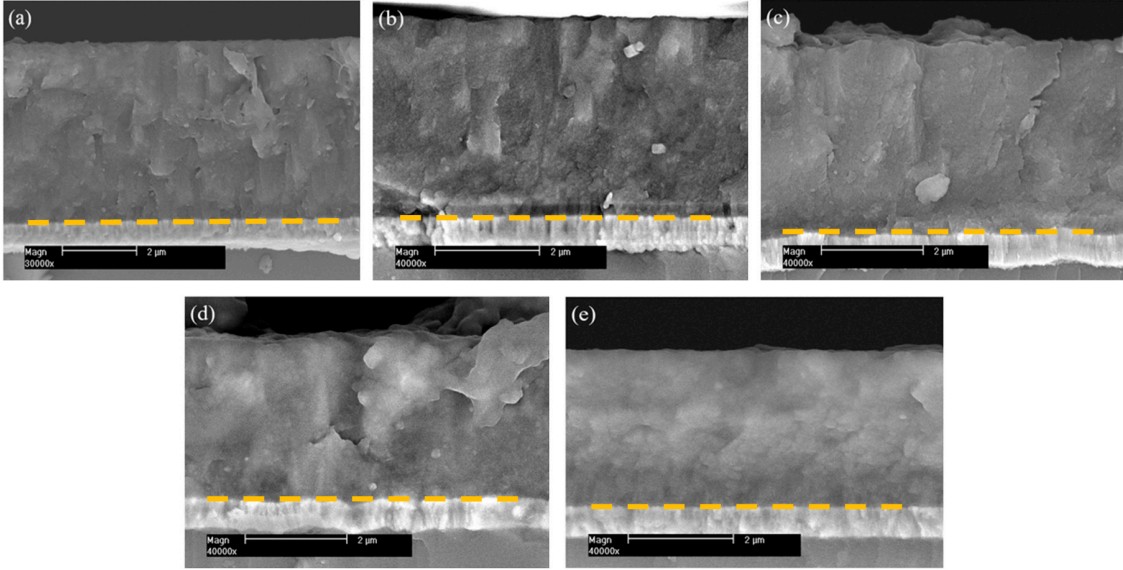

**Figure 5.** Cross-sectional morphologies of Cr-DLC films deposited at set values of: (**a**) 2%, (**b**) 5%, (**c**) 10%, (**d**) 15%, and (**e**) 20%. The start of the Cr-DLC films was marked by a yellow line in each figure.

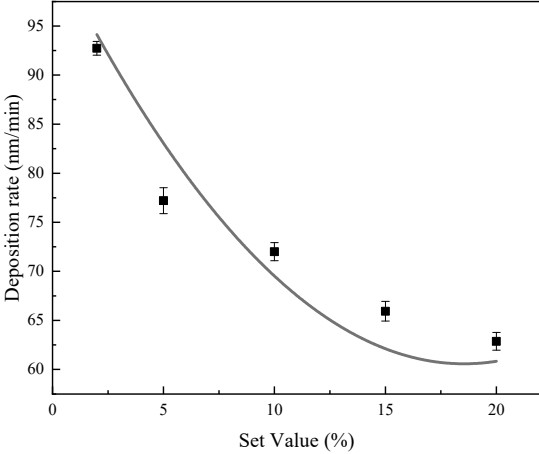

**Figure 6.** Deposition rates of Cr-DLC films as a function of the set value.

### 3.3. Effect of Set Values on the Microstructure

Raman spectra is one of the popular and effective nondestructive tools used to differentiate and acquire the bonding nature of DLC films. Figure 7 displays the Raman spectra between 1000 and 1800 cm$^{-1}$ for Cr-DLC films. In general, the Raman spectra can be fitted by two Gaussian distributions: one centered at about 1350 cm$^{-1}$, corresponding to the D peak of the disordered structure; and the another one centered at about 1580 cm$^{-1}$, related to the G peak of the graphite structure [25–27]. The integrated intensity ratio of the D and G band, $I_D/I_G$, can be correlated with the C–C $sp^3/sp^2$ bonding ratio [28]. Empirically, it is widely accepted that a larger $I_D/I_G$ ratio and higher G peak position means a lower C–C $sp^3$ content for non-hydrogenated DLC films [29,30]. The G peak positions slightly shift to a higher Raman frequency with an increasing set value. In Figure 8, the $I_D/I_G$ ratio increases from 1.94 to 2.76 when the set value increases from 2% to 20%, respectively. Nevertheless, for hydrogenated DLC films, this ratio influenced both C–C and C–H $sp^3$ bonding. Combined with the previous reports for the hydrogenated Cr-DLC [23,31,32], it was estimated that the $sp^3/sp^2$ ratio was approximately 0.4 for Cr-DLC films at the low set value, and this ratio decreases with the increase in set value. Such variations in the $sp^3/sp^2$ ratio with set value can be explained by the fact that a lower amount of $C_2H_2$ gas is introduced at a higher set value. As a result, more Cr and less hydrogen are incorporated into the Cr-DLC films. On the other hand, the overall Raman intensity of $sp^2$ bonds decrease with the increase of the set value, which means that the proportion of the DLC phase decreases.

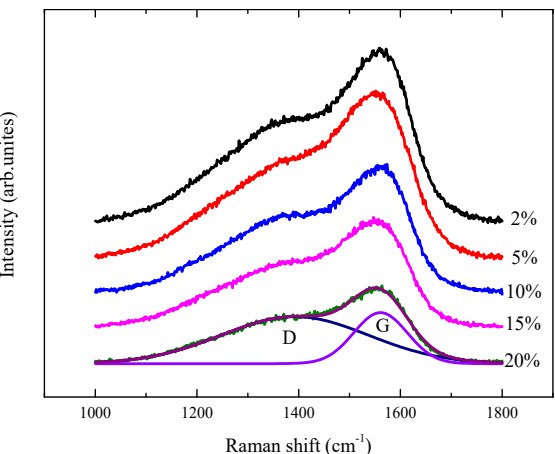

**Figure 7.** Raman spectra of the Cr-DLC films deposited at different set values.

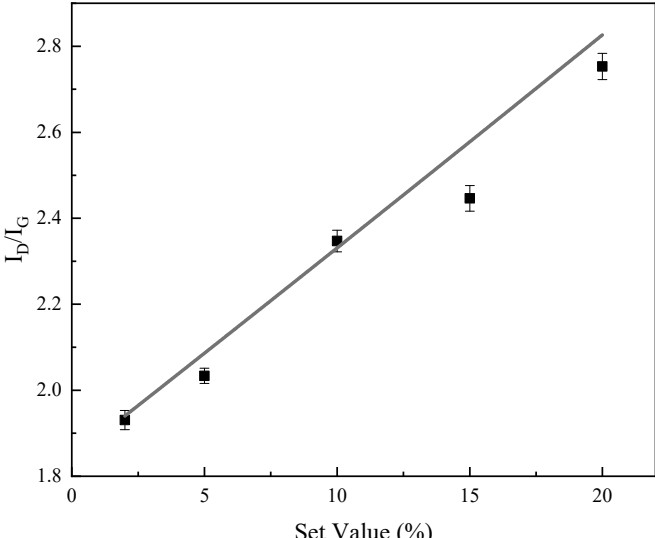

**Figure 8.** $I_D/I_G$ ratios of Cr-DLC films deposited at different set values.

### 3.4. Effect of Set Values on Mechanical Properties

The hardness and elastic modulus of the Cr-DLC films deposited at different set values are depicted in Figure 9. The hardness and elastic modulus increased monotonously from 5.72 to 8.02 GPa and from 80.19 to 139.63 GPa, respectively. According to the above Raman results, the proportion of the DLC phase tends to decrease when the set value increases, resulting in a decrease of hardness. However, with the increasing set value, more Cr atoms are incorporated into the Cr-DLC films, and hard chromium carbide nano-particles are formed and embedded in the DLC matrix, which significantly contributes to an increase in hardness and elastic modulus [8]. Based on the above analysis, we can conclude that hard chromium carbide phase was formed to offset the effect of the decrease of DLC phase that contributed to decreasing hardness and elastic modulus when set value increased from 2% to 20%. Overall, the hardness of the Cr-DLC films were quite low. Previous reports described similar phenomena for Cr-containing hydrogenated amorphous DLC films [33–35]. This is considered to be related to the higher hydrogen content in films. Excess hydrogen is bonded to carbon and leads to more C–H bonding in the Cr-DLC films and reduces the amount of C–C bonds forming the amorphous network, which reduces the strength of the amorphous phase. Furthermore, the lower deposition temperature of 75 °C is also another vital reason behind the relatively low hardness for Cr-DLC films.

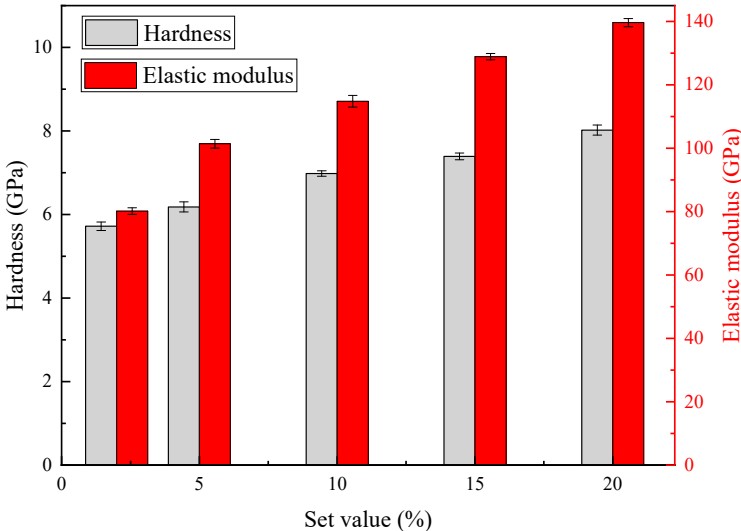

**Figure 9.** Hardness and Elastic modulus of the Cr-DLC films deposited at different set values.

Obtaining the high adhesion strength is one of the major technological challenges for the DLC films, and it plays an important role in the wear and corrosion resistance of the films. Figure 10 shows the critical loads ($LC_2$) of the Cr-DLC films deposited at different set values, and their critical loads are 15, 22, 18, 15 and 12N, respectively. The $LC_2$ is associated with the start of chipping failure extending from the arc tensile cracks, indicating adhesive failure between the coating and the substrate. The results indicate that the adhesion initially increased and then decreased when set value >5%. Generally speaking, the adhesion strength of films mainly depends on the film internal stress. In the initial stage, increasing set value meant more Cr content in the Cr-DLC film, resulting in a decrease in the proportion of DLC phase (confirmed by Raman spectra, see Figure 8), which was beneficial to the release of internal stress [36,37]. Therefore, the films possess a relatively lower internal stress when the set value increases up to 5%, which may improve the adhesion strength. Moreover, Choi et al. pointed out that metal doping reduced the directionality of bond, which led to a decrease in internal stress caused by bond angle distortion in the amorphous carbon network [38]. However, plenty of carbide phases will be generated with further increases of the set value. The Cr–C bond length is longer than the C–C bond length, causing an increase in internal stress [39,40]. As a result, the adhesion strength decreased as set value increased from 5% to 20%.

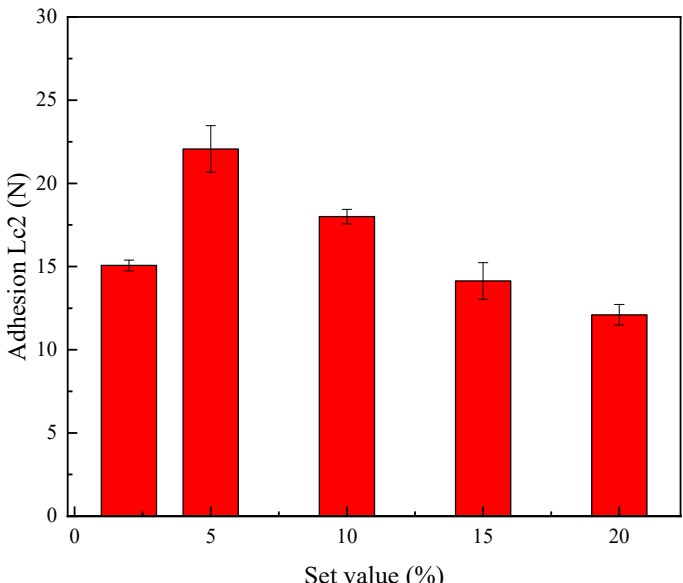

**Figure 10.** Adhesion of the Cr-DLC films deposited at different set values.

### 3.5. Effect of Set Values on Tribological Performance

Figure 11 depicts the friction coefficients evolution of the Cr-DLC films deposited at different set values, with the sliding time and corresponding surface profiles of the tracks after test. A smooth and indistinct track was observed for the films deposited at the set value of 2%, while the Cr-DLC films deposited at the set value > 5% presented deeper and broader wear tracks. Figure 12 shows the wear surfaces on the Cr-DLC films deposited at the set value of 2% and 20%. The wear track of the films deposited at a set value of 2% are larger than those deposited at a set value of 20%, which implied that the higher set value (meaning higher Cr content) deteriorated tribological performance of Cr-DLC films. Figure 13 shows the average friction coefficients, calculated after about 10 min of the sliding time, and the wear rates. The average friction coefficient and wear rate exhibited a relatively low value of 0.14 and $1.21 \times 10^{-9}$ mm$^3$/Nm at the set value of 2%, respectively. With the increase of set value, the average friction coefficient and wear rate both presented an increased tendency. The average friction coefficient and wear rate with set values of 5%, 10%, 15%, and 20% were 0.17 and $1.94 \times 10^{-9}$ mm$^3$/Nm, 0.18 and $2.16 \times 10^{-9}$ mm$^3$/Nm, 0.21 and $2.88 \times 10^{-9}$ mm$^3$/Nm, and 0.22 and $5.76 \times 10^{-9}$ mm$^3$/Nm, respectively. For the films deposited at low set values (low Cr content), the films displayed the characteristics of an

amorphous carbon structure, which led to the lower friction coefficient and wear rate. Nevertheless, higher Cr content led to the formation of a hard chromium carbide phase, and the amorphous Cr-DLC film was gradually transformed into carbide-rich film [41–43]. Thus, sliding between hard materials induced high shear strength, which in turn yielded high friction coefficients. These processes can yield abrasive wear, causing the deterioration of the tribological properties. On the other hand, the friction coefficients and wear properties were dominated by Cr metal at higher Cr content. During the sliding process, Cr metal interacted with the oxygen in the atmosphere, resulting in the formation of metal oxide [44,45], which increased the friction coefficient and the wear rate.

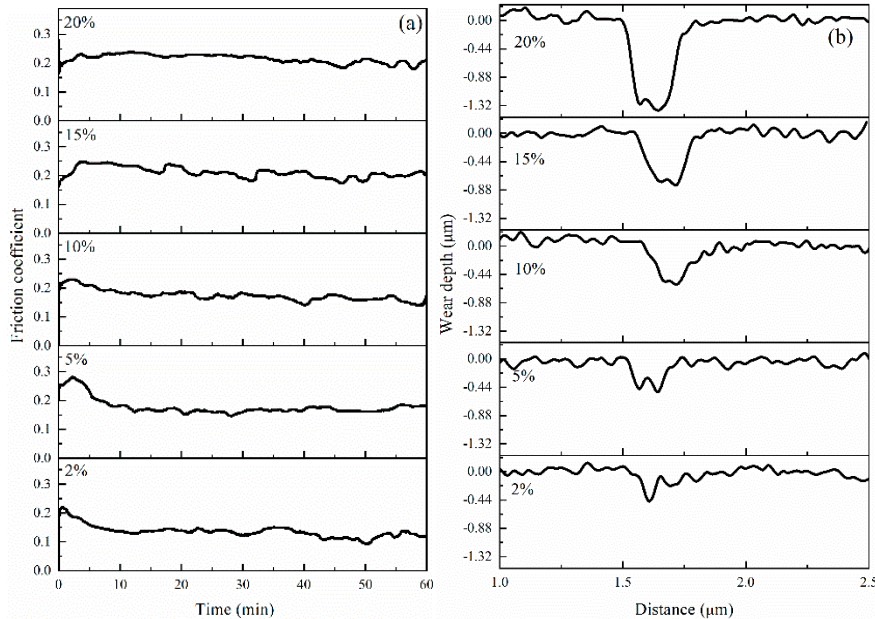

**Figure 11.** (**a**) Friction coefficient of the Cr-DLC films deposited at different set values and (**b**) corresponding surface profiles of the wear tracks after test.

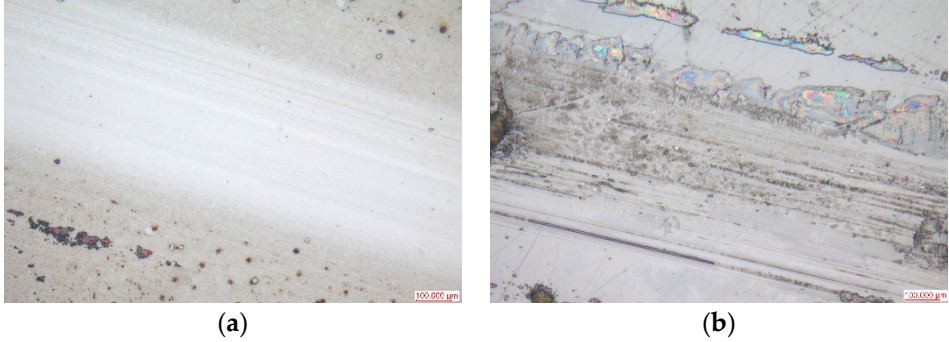

**Figure 12.** Wear tracks micrographs obtained by optical microscope of the Cr-DLC films deposited at set values of 2% (**a**) and 20% (**b**).

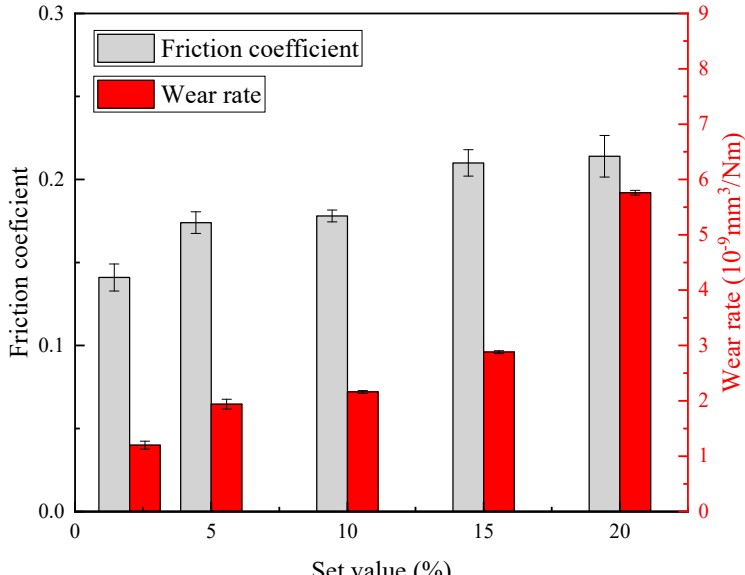

**Figure 13.** The average friction coefficient and wear rate of the Cr-DLC films deposited at different set values.

## 4. Conclusions

In summary, Cr-DLC films were synthesized by pulsed-dc magnetron sputter with a PEM system using a large-size industrial Cr target. The relationships between the Cr atom plasma emission intensity (set value), target voltage, pressure, and time suggest that the PEM system is an effective method for stabilizing the deposition process in the "transition" state. Compositional analysis showed that the Cr content increased from 1.96 to 17.21 at.%, and the set value increased from 2% to 20%. The Cr-DLC film deposition rate was as high as 92.7 nm/min at the set value of 2%, which is obviously higher than that for conventional hybrid magnetron sputtering. The adhesion strength increased initially and then reached its maximum values of 22 N at the set value of 5%. Raman results demonstrated that increasing set values promotes the graphitizing of Cr-DLC films. The hardness was enhanced with the increase of set value. In addition, it was observed that both the friction coefficient and wear rate increased with set value, and the lowest friction coefficient and wear rate were 0.14 and $1.2 \times 10^{-9}$ mm$^3$/Nm, respectively. Overall, the hardness values reported in this work were quite small; more experiments and deeper analysis are needed to reveal the mechanism behind it. On the other hand, there is a growing demand for over-sized targets (Length > 800 mm) in industrial applications. Due to the larger sputtered area, a single collimator is not enough to monitor the plasma emission intensity for the entire area. We are now designing multi-collimators and corresponding installation locations. More work is being conducted in this direction, and the results will be reported in due course.

**Author Contributions:** Conceptualization, methodology, supervision, G.L.; formal analysis, investigation, data curation, writing—original draft preparation, writing—review and editing, Y.X. (Yi Xu); project administration, funding acquisition, Y.X. (Yuan Xia). All authors have read and agreed to the published version of the manuscript.

**Funding:** This research was jointly found by National Nature Foundation of China (No. 51871230 & 51701229) and the Strategic Priority Research Program of the Chinese Academy of Sciences (No. XDB22040503).

**Conflicts of Interest:** The authors declare no conflict of interest.

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
