# Peer review of "Effect of Cr Atom Plasma Emission Intensity on the Characteristics of Cr-DLC Films Deposited by Pulsed-DC Magnetron Sputtering"

_coatings, doi:10.3390/coatings10070608_

Round 1

Reviewer 1 Report

1. The graph in Figures 6 and 8 should not be connected by measuring point lines, only to find a regression line fitted to the obtained data.
2. Relatively many literature items older than 5 years, add newer items
3. Improve the description of the state of the art in the Introduction and expand the discussion of the results obtained, e.g. shown in Fig. 10.
4. Add error bars in Figures 8, 10, 13.
5. Add information about the device for tribological tests.
6. The slip distance is calculated as the product of slip speed (240 mm / min) and time (60 min), why is it 5 mm instead of 14.4 m? If this is the path of a single tribotester stroke, this should be specified precisely.
7. The authors should provide the Archard formula for determining the wear rate value, and also summarize in Fig. 13 the value of volumetric wear.
8. The authors in point 3.1 compare methods of constant flow control mode vs. PEM system, but only in the range of emission intensity of Cr atom and C2H2 flow rate, but they did not show a comparison of both methods for mechanical (especially adhesion), tribological, structure and Raman spectra. It is recommended to complete this data. In this approach, the article is of little use.
9. Authors should present aspects of the novelty of their publication relative to publications [12,13].
10. In point 3.3 Authors should provide estimated contents of sp2 and sp3.
11. During the sliding process, Cr metal interact with the oxygen in atmosphere resulting into the formation of metal oxide, which increase the friction coefficient and decrease
(?) the wear rate - is this correct? This is not clear from Fig. 13.

Author Response

Dear reviewer:

Thanks for your comments about our manuscript “Effect of Cr atom plasma emission intensity on the characteristics of Cr-DLC films deposited by pulsed-dc magnetron sputtering” by G. Li et al. We have addressed all comments and suggestions raised by you and revised the manuscript accordingly. Please find attached the revised version of our manuscript, which we are resubmitting for consideration for publication in Coatings. All modifications in the manuscript are marked in RED.

Question 1: The graph in Figures 6 and 8 should not be connected by measuring point lines, only to find a regression line fitted to the obtained data.

Answer 1: Thanks for your advices. We have revised the Figures 6 and 8, in which a regression line is presented.

Question 2: Relatively many literature items older than 5 years, add newer items

Answer 2: Thanks for your advices. The older referred publications have been deleted and the newest publications have been referred in the paper.
Question 3: Improve the description of the state of the art in the Introduction and expand the discussion of the results obtained, e.g. shown in Fig. 10.

Answer 3: Thanks for your advices. We have improved the description of the state of the art in the Introduction and expand the discussion of all results.
Question 4: Add error bars in Figures 8, 10, 13.

Answer 4: Thanks for your advices. We have added the error bars in Figures 8, 10, 13.

Question 5: Add information about the device for tribological tests.

Answer 5: Thank you for checking the paper so carefully. In fact, adhesion strength, the friction coefficients, wear rates and profiles of the tracks were all evaluated by MFT-R4000, which is a multi-functional surface property testing equipment. In addition, we have added more information about the device for tribological tests in Line 111-117.

Question 6: The slip distance is calculated as the product of slip speed (240 mm / min) and time (60 min), why is it 5 mm instead of 14.4 m? If this is the path of a single tribometer stroke, this should be specified precisely.

Answer 6: Thank you for checking the paper so carefully. Here, 5 mm is the value of the track length. The total slip distance is 240 mm/min *60min=14.4 m. In order to avoid misunderstanding, the “slip distance” has been replaced with “track length” in Line 114.

Question 7: The authors should provide the Archard formula for determining the wear rate value, and also summarize in Fig. 13 the value of volumetric wear.

Answer 7: Thanks for your advices. We have provided the Archard formula for determining the wear rate value in the Experiment details (Line 1113-117) and summarized the value of volumetric wear rate (Line 247-250).

Question 8: The authors in point 3.1 compare methods of constant flow control mode vs. PEM system, but only in the range of emission intensity of Cr atom and C2H2 flow rate, but they did not show a comparison of both methods for mechanical (especially adhesion), tribological, structure and Raman spectra. It is recommended to complete this data. In this approach, the article is of little use.

Answer 8: Thank you for checking the paper so carefully. There are a large number of publications about the synthetization of the Cr-DLC by constant flow control mode. The mechanism of how the C2H2 gas flow rate affects the mechanical, adhesion, tribological, structure and Raman spectra has been widely investigated. Therefore, in this work, we pay more attention on the research of the relationships between Cr atom plasma emission intensity and the element concentration, cross-sectional morphology, deposition rate, microstructure, mechanical properties and tribological properties of Cr-DLC films under PEM system. In point 3.1, the aim of comparing methods of constant flow control mode vs. PEM system is to show the stable deposition process of the DLC films under PEM system.

Question 9: Authors should present aspects of the novelty of their publication relative to publications [12,13].

Answer 9: Thank you for checking the paper so carefully. The published literatures of the PEM system mainly focus the small sized target. For example, 200 mm-diameter Fe and Zr targets are chosen in publications [12] and [13], respectively. However, in this work, a large sized Cr target (131 mm×580 mm) is used to synthesize Cr-DLC films. Compared with the small sized targets, the large sized targets are more suitable for industrial production. In addition, the target poisoning and abnormal discharge (arcing) are more serious due to the larger sputtered area. Hence, our work can provide some referential materials for the industrialization of production.

Thanks for your advices, we have expanded the description of the introduction part (Line 56-62 ).

Question 10: In point 3.3 Authors should provide estimated contents of sp2 and sp3.

Answer 10: Thanks for your advices, we have provide estimated contents of sp2 and sp3 according to published papers (Line 188-192).

Question 11: During the sliding process, Cr metal interact with the oxygen in atmosphere resulting into the formation of metal oxide, which increase the friction coefficient and decrease (?) the wear rate - is this correct? This is not clear from Fig. 13.

Answer 11: We are very sorry for our incorrect writing. It is a slip of the pen. Thanks for your advices. With the increase of set value, the average friction coefficient and wear rate present an increased tendency. Now the descriptions have been revised in Line 259: During the sliding process, Cr metal interact with the oxygen in atmosphere resulting into the formation of metal oxide, which increase the friction coefficient and the wear rate.

Reviewer 2 Report

The manuscript “Effect of Cr atom plasma emission intensity on the 2 characterizations of Cr-DLC films deposited by 3 pulsed-dc magnetron sputtering” describes the use PEM as tool to better control the sputtering process and the properties of Cr-DLC thin films. It is generally well written and clear. Some clarification are needed before acceptance, especially in the description of the process.

Some suggestions about the points to be improved/clarified can be found bellow:

Title: please change the title, replacing “characterizations” with “characteristics”

Page 2 line 61: When defining the set value it is not very clear how the “percentage” is calculated. Is it percentage with respect to the emission in metallic mode? In this case more details should be given about the parameters that were kept constant: voltage, peak current or average power, gas pressure or total gas flow, flow of Ar gas etc. It is important to know all this details, since functioning in compound mode can be different depending on the control parameters

Table 1: please add the value of average power and/or voltage, depending on what was set constant.

Page 3 line 82, 300nm indentation seems very high. Please check the numbers and put it into relation with film thickness. Add the film thickness to the manuscript, only the interlayer thicknesses are given

In figure 3 the evolution of emission line intensity (relative to maximum intensity) and the gas flow are represented. From the abrupt change of emission in the constant flow rate mode, it results that the system was not in equilibrium when the acquisition started. It is normal to have an initial variation, towards an equilibrium state, which is around 6% in this case, corresponding to 53 sccm flow. In order to make a fair comparison between the two modes of operation, I suggest starting from same conditions, metallic mode for example, establish a moment in time, say t=0, when a) the gas flow is set to xx sccm, and b)set point is set to yy %, and then compare the two modes of operation over comparable periods of time. An even more accurate comparison would be if the set point of gas flow corresponds roughly to the one resulting from PEM control. In that respect, please change the scale of C2H2 flow to be the same for fig 3 a and b.

Page 4 line 100, please rephrase “deposition of Cr-DLC films starts, the set value of 10% is input in a very short time.” The meaning of “input” is confusing. The events of setting a parameter should be clearly marked in the graph with arrows, showing at what point in time the change was made.

Page 4 line 111, please specify Cr-DLC films , instead of DLC films

 In fig 4 it is seen that the voltage is changing when the PEM is regulating the gas flow. It results then that the power is also changing, since the current is kept constant. Please clarify these aspects, since it is important to know what parameter is kept constant.

Page 5 line 126, please rephrase “number of oxygens”

Page 5 line 127is said that the presence of oxygen atoms is the reason of the water contamination. The formulation confuses cause and effect, the contamination of the walls is the cause for oxygen atoms presence. Please clarify

Page 5. In the discussion about the evolution of the deposition rate please add information about the variation of power input, since this is in important parameter that influences the deposition rate. When the setpoint increases, the amount of sputtered Cr increases and the deposition rate should increase.

Section 3.4, the hardness values reported are quite small. Please refer and compare to results from literature to support these findings. The large penetration depth is a possible reason for underestimating the hardness.

Please add some error bars in figure 10.

Author Response

Dear reviewer:

Thanks for your comments about our manuscript “Effect of Cr atom plasma emission intensity on the characteristics of Cr-DLC films deposited by pulsed-dc magnetron sputtering” by G. Li et al. We have addressed all comments and suggestions raised by you and revised the manuscript accordingly. Please find attached the revised version of our manuscript, which we are resubmitting for consideration for publication in Coatings. All modifications in the manuscript are marked in RED.

Question 1: Title: please change the title, replacing “characterizations” with “characteristics”

Answer 1: Thanks for your advices. We have replaced “characterizations” with “characteristics” in the title.

Question 2: Page 2 line 61: When defining the set value it is not very clear how the “percentage” is calculated. Is it percentage with respect to the emission in metallic mode? In this case more details should be given about the parameters that were kept constant: voltage, peak current or average power, gas pressure or total gas flow, flow of Ar gas etc. It is important to know all this details, since functioning in compound mode can be different depending on the control parameters.

Answer 2: In this work, the set value of 100% and 0% were defined as the pure metallic deposition and fully poisoning deposition, respectively. The set value between 0% and 100% represents the percentage of the reduced Cr atom plasma emission intensity.

You can find the constant parameters in the Table 1. The pulse current is 5 A and the Ar gas pressure is 0.5 Pa (150 sccm). In addition, thanks for your advices. We have repeated all experiments at value of 2%, 5%, 10%, 15% and 20% and recorded the target voltages and deposition pressure evolutions during the films’ deposition, and the experiment results are shown in Fig.4.

Question 3: Table 1: please add the value of average power and/or voltage, depending on what was set constant.

Answer 3: Thanks for your advices. The target voltage evolutions at different set values are presented in Fig.4. As the pulse current is kept constant (5A), the average power has the same variation tendency to the target voltage and the average powers at stable deposition state are 1.44kW at 2%, 1.42kW at 5%, 1.38kW at 10%, 1.36kW at 15% and 1.33kW at 20%. (see table 1 and Line 140-141)

Question 4: Page 3 line 82, 300nm indentation seems very high. Please check the numbers and put it into relation with film thickness. Add the film thickness to the manuscript, only the interlayer thicknesses are given.

Answer 4: Thanks for your advices. According to the images in Figure 5, it is observed that the film thicknesses at 2%, 5%, 10%, 15% and 20% are 4.6μm, 3.8μm,3.6μm,3.3μm and 3.1μm, respectively. The data have been added in Line 163-164. Generally speaking, the substrate effect can be avoided when the penetration depth of the indenter is controlled to be less than one-tenth of the deposited film's thickness. In my opinion, the 300 nm indentation is suitable.

Question 5: In figure 3 the evolution of emission line intensity (relative to maximum intensity) and the gas flow are represented. From the abrupt change of emission in the constant flow rate mode, it results that the system was not in equilibrium when the acquisition started. It is normal to have an initial variation, towards an equilibrium state, which is around 6% in this case, corresponding to 53 sccm flow. In order to make a fair comparison between the two modes of operation, I suggest starting from same conditions, metallic mode for example, establish a moment in time, say t=0, when a) the gas flow is set to xx sccm, and b)set point is set to yy %, and then compare the two modes of operation over comparable periods of time. An even more accurate comparison would be if the set point of gas flow corresponds roughly to the one resulting from PEM control. In that respect, please change the scale of C2H2 flow to be the same for fig 3 a and b.

Answer 5: In Fig 3, the two modes are operated at the same starting conditions. At first, it should be noted that the time = 0 s is the beginning of the experiments. Secondly, the records have a delay time of 30 millisecond and the data of the set values are automatically by computer recorded after the C2H2 flow rate is introduced into the chamber. Under constant flow control mode, the C2H2 flow rate is kept constant (53 sccm). However, under PEM system at a set value of 10%, the C2H2 flow rate is about 70 sccm at the beginning of the experiment. Therefore, the C2H2 flow rates at the time=0 s for constant flow control mode and PEM system are different and then the corresponding set values are different.

In addition, thanks for your advices. The scale of C2H2 flow is the same for Fig.2 (a) and (b) now.

Question 6: Page 4 line 100, please rephrase “deposition of Cr-DLC films starts, the set value of 10% is input in a very short time.” The meaning of “input” is confusing. The events of setting a parameter should be clearly marked in the graph with arrows, showing at what point in time the change was made.

Answer 6: Thanks for your advices. In order to avoid the misunderstanding, we revised the descriptions about the set value at the beginning of the deposition: As the deposition of Cr-DLC films starts, the set value of Cr decreases quickly to the set value and then keep stable (Line 128-130). In addition, a detailed description about the event of setting a parameter have been added in the caption of the Fig 3.

Question 7: Page 4 line 111, please specify Cr-DLC films, instead of DLC films

Answer 7Thanks for your advices. We have replaced “DLC” with “Cr-DLC” Line 142.

Question 8: In fig 4 it is seen that the voltage is changing when the PEM is regulating the gas flow. It results then that the power is also changing, since the current is kept constant. Please clarify these aspects, since it is important to know what parameter is kept constant.

Answer 8: We appreciate your reading the article so carefully. In this work, a pulsed-dc magnetron sputtering is used to synthesize Cr-DLC films. The pulse current, pulse frequency and duty ratio are kept constant. These deposition parameters are summarized in Table 1. In order to avoid misunderstanding, we emphasized these aspects both in 2.1 deposition (Line 78-79 ) and 3.1 stability comparison (Line 139-141).

Question 9: Page 5 line 126, please rephrase “number of oxygens”. Page 5 line 127 is said that the presence of oxygen atoms is the reason of the water contamination. The formulation confuses cause and effect, the contamination of the walls is the cause for oxygen atoms presence. Please clarify

Answer 9: We are very sorry for our incorrect writing. We have revised the descriptions Line 158-159: In addition, there is a small amount of oxygen in the films, which may be caused by the contamination from the residual vacuum.

Question 10: Page 5. In the discussion about the evolution of the deposition rate please add information about the variation of power input, since this is in important parameter that influences the deposition rate. When the setpoint increases, the amount of sputtered Cr increases and the deposition rate should increase.

Answer 10: The target voltage evolutions at different set values are presented in Fig.4. As the pulse current is kept constant (5A), the average power has the same variation tendency to the target voltage. In table 1 and Line 140-141, we have added the descriptions about the average power. According to the EDS results, the amount of sputtered Cr increases with the increase of set value. However, the C2H2 gas flow rate is reduced. The ionization energy for Ar and C2H2 are 15.75 eV and 11.403 eV, respectively. Therefore, C2H2 are more apt to be ionized than Ar. In other word, C element is more active during the deposition. Carbon-containing particles are more likely to settle down and form carbon-based films on the substrate. As a result, the deposition rates decrease with the increase of set value.

Question 11: Section 3.4, the hardness values reported are quite small. Please refer and compare to results from literature to support these findings. The large penetration depth is a possible reason for underestimating the hardness.??

Answer 11: The hardness in this work is quite small. In my opinion, excess hydrogen in Cr-DLC films may be responsible for the small hardness. Excess hydrogen is bonded to carbon and leads to more C–H bonding in the Cr-DLC film and reduces the amount of C–C bonds forming the amorphous network, and in this way reduces the strength of the amorphous phase. What’s more, the lower deposition temperature of 75℃ is also another vital reason behind the quite low hardness for Cr-DLC films.

Thanks for your advice. We have expanded the discussion in section 3.4 and more publications are referred to support these findings (Line 209-215).

Generally speaking, the substrate effect can be avoided when the penetration depth of the indenter is controlled to be less than one-tenth of the deposited film's thickness. In my opinion, the 300nm indentation is suitable.

Question 12: Please add some error bars in figure 10.

Answer 12: Thanks for your advices. The error bars have been added in Fig.10.

Reviewer 3 Report

The work presented in this article deals with the synthesis of Cr-doped hydrogenated DLC coatings using pulsed DC and by employing PEM setup. The structural, mechanical and tribological properties have been investigated with regards to the set-value of Cr atom emission from the plasma discharge.

The article is well written, the experiment planning and execution is sound and the results are presented in a structured way. The article should be accepted for publication with minor revision.

Following are my general and specific comments and questions for the authors to improve the quality of the manuscript.

Language

While the language quality is good, there are still some spelling and grammatical mistakes. I strongly recommend the authors to proof-read the manuscript before submitting the revised version.

Introduction

  • The authors mention the advantage of the PEM system by referring to various film deposition processes such as CrNx, TiO2 and AZO where PEM method was employed. Has anyone else used the PEM setup for Cr doped DLC:H using C2H2? If yes, please refer that work also.
  • Also, in the last paragraph the authors mention that a PEM system was developed in this work: by developing do the authors mean they have developed something new or is it just the standard PEM setup that was simply employed in this work? Please clarify and change the text if needed.
  • It will be good if the authors can show the readers what is the big picture here? This is clearly stated and referred by other works that a PEM system is useful in providing process stability which is needed in all deposition processes. The results show that the PEM system also works very well for this particular process and coating but is there any challenge in, for example, scaling this setup to an industrial scale? How can this setup benefit large scale production of these coatings? Are the film properties achieved in this work extraordinary and previously unachieved without a PEM setup? Please specify the future direction and scope of this work.

Experimental Details

  • What was the thickness of the DLC coating? Were the thicknesses different for coatings deposited on Si and steel? Please specify.
  • Was the substrate holder stationary or rotating?
  • Please specify which pulsed DC power supply and which power supply was used for the substrate bias.
  • Was the DC bias (with the same voltage value of -75) used also during the Cr and CrN interlayer depositions?
  • What were the target power densities used for the interlayers and for the DLC layer? Please mention them.

Results and Discussion

  • Given that the authors mention about Nanocomposite DLC structure with metal doping and its benefit for superior mechanical properties, it comes as a surprise that the authors do not show any TEM results for ascertain this fact. Why was such an analysis not performed?
  • Looking at microstructure and hardness values in Fig. 5 and Fig. 9, a discrepancy is observed. The films with 10% set-value appears to be the most dense whereas the hardness is the highest for the highest set-value i.e. 20%. What could be the reason for this behavior? Also, did the authors measure the film density for example using x-ray reflectivity?
  • What is the state-of-the-art of the Cr doped DLC:H coatings? How do the mechanical, tribological and structural properties of the film presented in this work compare to the literature? Please refer other works.
  • Overall, the hardness values of the films (below 10 GPa) are quite low. Are these values in line with the aim and the expectations?
  • Discussing the Raman results, the authors conclude that the overall DLC phase decreases with an increase in the set-value, yet the hardness is increasing with the set-value; what this increase in hardness stem from?
  • The lowest friction coefficient of 0.14 seems to be still higher than what could be achieved using various hydrogenated DLCs. How could this be improved?

Conclusions

  • While summarizing the work and commenting on the important findings, it is also important to show the future direction and benefit or simply what should be the next step in this particular direction of the research. Please mention the big picture of this work in the introduction and also support it through your results in the conclusion.

Author Response

Dear reviewer:

Thanks for your comments about our manuscript “Effect of Cr atom plasma emission intensity on the characteristics of Cr-DLC films deposited by pulsed-dc magnetron sputtering” by G. Li et al. We have addressed all comments and suggestions raised by you and revised the manuscript accordingly. Please find attached the revised version of our manuscript, which we are resubmitting for consideration for publication in Coatings. All modifications in the manuscript are marked in RED.

Language

Question 1: While the language quality is good, there are still some spelling and grammatical mistakes. I strongly recommend the authors to proof-read the manuscript before submitting the revised version.
Answer 1: Thanks for your advices. We have revised the manuscript carefully and tried to avoid any grammar or syntax errors. In addition, we have asked several colleagues who are skilled authors of English language papers to check the English. We believe that the language is now acceptable for the review process.

Introduction

Question 2: The authors mention the advantage of the PEM system by referring to various film deposition processes such as CrNx, TiO2 and AZO where PEM method was employed. Has anyone else used the PEM setup for Cr doped DLC:H using C2H2? If yes, please refer that work also.

Answer 2: Thanks for your advices. In fact, some researchers have used the PEM system to synthesize Ti-containing and W-containing hydrogenated DLC films by HiPIMS in recent years. The corresponding publications have been referred in Line L48-54.

Question 3: Also, in the last paragraph the authors mention that a PEM system was developed in this work: by developing do the authors mean they have developed something new or is it just the standard PEM setup that was simply employed in this work? Please clarify and change the text if needed.

Answer 3: We appreciate your reading the article so carefully. It is just the standard PEM setup that is employed in this work. In order to avoid the misunderstanding, we have revised the descriptions in Line 63: In this work, a pulsed-dc magnetron sputtering with PEM system was applied to…

Question 4: It will be good if the authors can show the readers what is the big picture here? This is clearly stated and referred by other works that a PEM system is useful in providing process stability which is needed in all deposition processes. The results show that the PEM system also works very well for this particular process and coating but is there any challenge in, for example, scaling this setup to an industrial scale? How can this setup benefit large scale production of these coatings? Are the film properties achieved in this work extraordinary and previously unachieved without a PEM setup? Please specify the future direction and scope of this work.

Answer 5: We appreciate your reading the article so carefully. In this work, the vacuum chamber mounted vertically with rectangular Cr (99.99% purity) target (131 mm×580 mm) and this target size is industrial scale. In fact, we have successfully applied the PEM system to synthesize DLC and Al2O3 films as anti-corrosion films for the NdFeB magnet and achieved mass production. Unstable and uncontrolled deposition process (arcing discharge and significant decrease in deposition rate) are observed without PEM system. Under PEM system, the stable deposition processes of the Cr-DLC films are achieved without anormal discharge. And the same time, when the set value is 2%, the Cr-DLC film deposition rate is as high as 92.7nm/min. The deposition of these Cr-DLC films under PEM system presents a higher film deposition rate than that for conventional hybrid magnetron sputter deposition processes.

The future direction and scope of this work: Due to the lower hardness, we need more experiments and measurements (such as TEM) to reveal the mechanism behind it. On the other hand, there is a growing demand for over-sized targets in industry. Due to the larger sputtered area, single collimator is not enough to monitor the plasma emission intensity for the entire area. Now we are designing multi- collimators and corresponding installation locations. More work is being conducted in this direction and the results will be reported in due course.

Thanks for your advices, we have expanded the descriptions of the conclusion and added the future direction and scope of this work.

Experimental Details

Question 6: What was the thickness of the DLC coating? Were the thicknesses different for coatings deposited on Si and steel? Please specify.

Answer 6: Thanks for your advices. According to the images in Figure 5, it is observed that the film thicknesses at 2%, 5%, 10%, 15% and 20% are 4.6μm, 3.8μm,3.6μm,3.3μm and 3.1μm, respectively. The data have been added in Line 163-164. In addition, there are no difference in thickness for films deposited on Si and steel in this work.

Question 7: Was the substrate holder stationary or rotating?

Answer 7: During the deposition process, the substrate holder is stationary. Thanks for your advices. The corresponding descriptions have been added in Line 99.

Question 8: Please specify which pulsed DC power supply and which power supply was used for the substrate bias.

Answer 8: Thanks for your advices. During the deposition process, a pulsed DC power (MSP-20D, Pulsetech) is used to power the Cr target, while a DC power (MSP-50D, Pulsetech) is applied to the substrate. And the corresponding descriptions have been added in experiment details (Line 78-80). The setting parameters have been summarized in table 1.

Question 9: Was the DC bias (with the same voltage value of -75) used also during the Cr and CrN interlayer depositions? What were the target power densities used for the interlayers and for the DLC layer? Please mention them.

Answer 10: Thanks for your advices. The DC bias is -75 V during the Cr and CrN interlayer depositions. The corresponding descriptions have been revised in Line 93-96: Then the deposition of layered interlayer containing of Cr (150nm) and CrN (500nm) were deposited at a bias of -75 V in sequence at a Ar gas deposition pressure of 0.5 Pa in a pulsed DC mode to power the target to 5 A with a frequency of 40kHz and a duty ratio of 80%. The CrN films were deposited under PEM system and the set value was fixed at 60%.

The related descriptions have been added in Line 93-96.

For the deposition process, the target voltage evolutions at different set values are presented in Fig.4. As the pulse current is kept constant (5A), the average power has the same variation tendency to the target voltage. In table 1 and Line 93-96, we have added the descriptions about the average powers at different set values at the stable deposition process.

Results and Discussion

Question 11: Given that the authors mention about Nanocomposite DLC structure with metal doping and its benefit for superior mechanical properties, it comes as a surprise that the authors do not show any TEM results for ascertain this fact. Why was such an analysis not performed?

Answer 11: We appreciate your reading the article so carefully. In the introduction part, we mention: By the doping of metals, the two-dimensional array of nanoclusters within the DLC matrix or an atomic-scale composite can be formed, and these nanostructures are responsible for the desirable mechanical and thermal properties. These results have been confirmed by plenty of researchers. Here, we just want to express that superior mechanical properties of Me-DLC is the result of the nanocomposite structure. The main purpose of this work is to evaluate the feasibility of PEM system in synthesizing Cr-DLC films using a large-size industrial Cr target and C2H2 as feeding gas, and investigated the effect of Cr atom plasma emission intensity on the microstructure and properties of the Cr-DLC films.

Thanks for your advices. The TEM measurement is very important tool for deeper analysis of the films’ microstructure. Due to the lower hardness, we need more experiments and measurements (such as TEM) to reveal the mechanism behind it. More work is being conducted in this direction and the results will be reported in due course.

Question 12: Looking at microstructure and hardness values in Fig. 5 and Fig. 9, a discrepancy is observed. The films with 10% set-value appears to be the most dense whereas the hardness is the highest for the highest set-value i.e. 20%. What could be the reason for this behavior? Also, did the authors measure the film density for example using x-ray reflectivity?

Answer 12: We do not measure the film density by XRR. From my point of view, there are no significant difference in densification for all samples according to the Fig.5. At the same time, the preferred orientation, grain size, densification, and residual stress of the films all have a key influence on the hardness. In the section 3.4, we have discussed the evolution of the hardness with increasing of the set value: with the increasing set value, more Cr atoms are incorporated into the Cr-DLC films, and then hard chromium carbide nano-particles are formed and embedded in the DLC matrix, which significantly contribute to an increase in hardness and elastic modulus. You can find more detailed discussions in section 3.4.

Question 13: What is the state-of-the-art of the Cr doped DLC:H coatings? How do the mechanical, tribological and structural properties of the film presented in this work compare to the literature? Please refer other works.

Answer 13: The high power pulse magnetron sputtering (HiPIMS) is the state-of-the-art technology for synthesizing Cr doped DLC:H films. The related publications have been refereed in the introduction part and 3.2 section. Compare with the films deposited by HiPIMS technology, the Cr-DLC films in this work have significant advantage in the stable deposition process and deposition rate, but have some disadvantage in the hardness and friction performance.

Question 14: Overall, the hardness values of the films (below 10 GPa) are quite low. Are these values in line with the aim and the expectations? Discussing the Raman results, the authors conclude that the overall DLC phase decreases with an increase in the set-value, yet the hardness is increasing with the set-value; what this increase in hardness stem from?

Answer 14: The hardness in this work is quite small. These values are not in line with aim and the expectations. In my opinion, excess hydrogen in Cr-DLC films is responsible for the small hardness. Excess hydrogen is bonded to carbon and leads to more C–H bonding in the Cr-DLC film and reduces the amount of C–C bonds forming the amorphous network, and in this way reduces the strength of the amorphous phase. We have expanded the discussion in section 3.4 and more publications are referred to support this findings (Line). What’s more, the lower deposition temperature of 75℃ is also another vital reason behind the quite low hardness for Cr-DLC films.

In the section 3.4, we have discussed the evolution of the hardness with increasing of the set value: with the increasing set value, more Cr atoms are incorporated into the Cr-DLC films, and then hard chromium carbide nano-particles are formed and embedded in the DLC matrix, which significantly contribute to an increase in hardness and elastic modulus. You can find more detailed discussions in Line.

Question 15: The lowest friction coefficient of 0.14 seems to be still higher than what could be achieved using various hydrogenated DLCs. How could this be improved?

Answer 15: For hydrogenated DLC films, the friction coefficient is related to the application environment in great extent. At dry N2 or high vacuum, a super-low friction coefficient can be acquired. But, at the air atmosphere or humid environment, the friction coefficient is higher. In this work, the friction coefficients were evaluated at a relative humidity of approximately 22% RH and temperature of 23 °C. In my opinion, the lowest friction coefficient of 0.14 is normal for hydrogenated DLC films. Similar results have been observed for Cr- containing (references [23], [33] and [41])and other metal-containing (reference [8],[17] and [22]) amorphous hydrogenated carbon films. On the other hand, we are trying to develop new methods to reduce the friction coefficient. For example, we plan to synthesize Cr-DLC films by HiPIMS. The HiPIMS technique is capable of offering not only highly dense plasma, which has been found to allow for an intense bombardment of high-energy particle and an increased adatom mobility on the growing films. Hence, the HiPIMS-deposited films exhibit lower surface roughness and lower friction coefficient. In addition, we care considering a containing element change. The Si, S and Ti are the target, and co-doping by multi-elements are also considered.

Conclusions

Question 16: While summarizing the work and commenting on the important findings, it is also important to show the future direction and benefit or simply what should be the next step in this particular direction of the research. Please mention the big picture of this work in the introduction and also support it through your results in the conclusion.

Question 16: Thanks for your advices. We have revised the conclusion and introduced next step in this particular direction of the research

Round 2

Reviewer 1 Report

I have no further commentsI have no further comments